# Açai supplementation (*Euterpe oleracea Mart*.) attenuates cardiac remodeling after myocardial infarction in rats through different mechanistic pathways

Amanda Menezes Figueiredo[1]*, Ana Carolina Cardoso[1], Bruna Leticia Buzati Pereira[1], Renata Aparecida Candido Silva[1], Andrea Freitas Goncalves Della Ripa[1], Tatiana Fernanda Bachiega Pinelli[1], Bruna Camargo Oliveira[1], Bruna Paola Murino Rafacho[1], Larissa Lumi Watanabe Ishikawa[2], Paula Schmidt Azevedo[1], Katashi Okoshi[1], Ana Angelica Henrique Fernandes[2], Leonardo Antonio Mamede Zornoff[1], Marcos Ferreira Minicucci[1], Bertha Furlan Polegato[1], Sergio Alberto Rupp Paiva[1]

1 Internal Medicine Department, Botucatu Medical School, Universidade Estadual Paulista (UNESP), Botucatu, São Paulo, Brazil, 2 Chemistry and Biochemistry Department, Institute of Biosciences, Universidade Estadual Paulista (UNESP), Botucatu, São Paulo, Brazil

* amanda.mfigueiredo@yahoo.com.br

**Data Availability Statement:** All files are available from the UNESP database https://repositorio.unesp.br/bitstream/handle/11449/136446/

## Abstract

Myocardial infarction has a high mortality rate worldwide. Therefore, clinical intervention in cardiac remodeling after myocardial infarction is essential. Açai pulp is a natural product and has been considered a functional food because of its antioxidant/anti-inflammatory properties. The aim of the present study was to analyze the effect of açai pulp supplementation on cardiac remodeling after myocardial infarction in rats. After 7 days of surgery, male Wistar rats were assigned to six groups: sham animals fed standard chow (SA0, n = 14), fed standard chow with 2% açai pulp (SA2, n = 12) and fed standard chow with 5% açai pulp (SA5, n = 14), infarcted animals fed standard chow (IA0, n = 12), fed standard chow with 2% açai pulp (IA2, n = 12), and fed standard chow with 5% açai pulp (IA5, n = 12). After 3 months of supplementation, echocardiography and euthanasia were performed. Açai pulp supplementation, after myocardial infarction, improved energy metabolism, attenuated oxidative stress (lower concentration of malondialdehyde, P = 0.023; dose-dependent effect), modulated the inflammatory process (lower concentration of interleukin-10, P<0.001; dose-dependent effect) and decreased the deposit of collagen (lower percentage of interstitial collagen fraction, P<0.001; dose-dependent effect). In conclusion, açai pulp supplementation attenuated cardiac remodeling after myocardial infarction in rats. Also, different doses of açai pulp supplementation have dose-dependent effects on cardiac remodeling.

figueiredo_am_me_bot.pdf?sequence=
3&isAllowed=y.

**Funding:** This research was supported in part by Coordenação de Aperfeiçoamento Pessoal de Nível Superior (CAPES), Brazil. The funding source had no involvement in study design, in the collection, analysis and interpretation of data, in the writing of the report, and in the decision to submit the article for publication. There was no additional external funding received for this study.

**Competing interests:** The authors have declared that no competing interests exist.

## Introduction

Cardiovascular disease (CVD) is a major cause of morbidity and mortality worldwide. Myocardial infarction (MI) is associated with a higher mortality rate than other CVDs [1]. MI is defined as a focus of necrosis resulting from low tissue perfusion, with signs and symptoms consequent to cardiac cell death [2]. Cellular and molecular alterations initiate a cascade of intracellular signaling, which increases inflammation, oxidative stress, apoptosis, and changes in cardiac energy metabolism [3, 4]. Initially, cardiac remodeling is a relevant factor in the progression of CVD because it plays a fundamental role in the pathophysiology of ventricular dysfunction [5, 6].

Therefore, inflammatory processes and oxidative stress are potential targets for attenuating cardiac remodeling and reducing mortality after MI. In this context, specific foods with high antioxidant properties have been used as sources of cardioprotective compounds [7–9]. Açai (*Euterpe oleracea* Mart.) is a plant in the Arecaceae family and a palm fruit native to the Amazon region of Brazil. Açai seeds have been used in animal foods, plantations and home gardens. Açai pulp is a functional food, consumed in energy drinks, ice cream, juice, and pharmaceutical products; and used in cosmetics [10, 11]. Notably, the consumption of açai pulp has been increasing worldwide.

Açai seed and pulp are rich in several phytochemicals but differ in the degree of richness. Açai seeds contain 65% fiber, 5% protein, 2% lipids, 2% mineral salts, and 28.3% polyphenols (catechin, epicatechin, and polymeric and oligomeric proanthocyanidins) [12]. Açai pulp contains 48% lipids, 25% total sugars, 13% protein, a small amount of fiber, vitamins (A, B1, B2, B3, C, and E), and 25.5% polyphenols (predominantly cyanidin 3-glucoside and cyanidin 3-rutinoside) [13, 14].

In experimental models, açai seed supplementation was performed using hydroalcoholic seed extract (ASE) [15, 16]. Because the seeds are not edible, supplementation has been achieved using açai capsules in clinical studies. Additionally, açai pulp is edible and more attractive than the seeds because of its sensorial characteristics (i.e., appearance, texture, and taste). These characteristics remain unchanged when the pulp is pasteurized and frozen for storage [17]. Thus, açai pulp can be easily used in clinical and experimental studies [18–20].

Studies have shown in experimental models that ASE supplementation has anti-inflammatory [15, 16] and antioxidative activities [21, 22]. This anti-inflammatory activity was also observed in açai pulp supplementation by decreased transcription of nuclear factor kappa B (NF-κB) in the brain [23]. In addition, supplementation with açai pulp decreased the concentration of interferon-gamma (IFN-γ) in individuals with metabolic syndrome [20].

Regarding antioxidative activity, supplementation with açai pulp decreased nitric oxide in microglial cells [24]. Studies performed in different clinical situations reported that açai pulp supplementation increased total antioxidant capacity (TAC) and attenuated exercise-induced muscle damage in junior athletes [25], increased catalase and TAC, and reduced the production of reactive species (RS) and malondialdehyde (MDA) in healthy volunteers [19, 26].

Açai pulp supplementation has also promoted benefits in the heart; for instance, a study of doxorubicin-induced cardiotoxicity in rats found attenuation of this toxicity by decreased oxidative damage (increased glutathione peroxidase and decreased lipid hydroperoxide concentration). Furthermore, açai pulp supplementation improved energy metabolism in the heart (increased *β*-hydroxyacyl-CoA dehydrogenase, decreased phosphofructokinase; and increased citrate synthase, complex II, and adenosine triphosphate [ATP] synthase enzymatic activities) [27].

In a search of electronic libraries (PubMed, Scopus, and Cochrane) for information on MI and açai supplementation, two articles were found. An experimental model of coronary artery

ligation [28], and a global ischemia-reperfusion model in rats [29]. In the first study, açai seed supplementation was performed for 4 weeks, and the rats received 100 mg/kg/day of ASE. The authors observed attenuation in hypertrophy (decreased heart weight), fibrosis (decreased collagen deposition in the left ventricle), and cardiac dysfunction (alterations in arterial and left ventricular pressure) [28]. In the second study, the authors supplemented 5% of açai pulp for 6 weeks and observed an improvement in energy metabolism and attenuation of oxidative stress [29]. Thus far, whether açai pulp supplementation affects cardiac remodeling after MI remains unknown. Although açai pulp has antioxidant and anti-inflammatory activities, its chemical composition differs from that of the seed, as afore mentioned.

Therefore, the aim of this study was to evaluate the effect of açai pulp supplementation in rat chow on cardiac remodeling after MI by modifying oxidative stress, energy metabolism, and inflammatory pathways.

## Materials and methods

All experiments and procedures were performed in accordance with the National Institute of Health's Guide for the Care and Use of Laboratory Animals and with the Ethical Principles in Animal Experimentation adopted by the Brazilian College of Animal Experimentation. The study protocol (n˚1066/2013) was submitted and approved by the Botucatu Medical School Animal Research Ethics Committee.

Male Wistar rats weighing from 200 to 250 g. Animals were subjected to MI according to a method previously described [2, 30] or sham surgery. After surgery, rats were housed in individual cages, in a temperature-controlled room (24˚C) with a 12-hour light/12-hour dark cycle. Water was supplied *ad libitum*, and food was controlled.

After the initial echocardiographic exam, animals were allocated into six groups: (1) sham animals fed standard chow (SA0, n = 14); (2) sham animals fed standard chow with 2% açai pulp (SA2, n = 12); (3) sham animals fed standard chow with 5% açai pulp (SA5, n = 14); (4) infarcted animals fed standard chow (IA0, n = 12); (5) infarcted animals fed standard chow with 2% açai pulp (IA2, n = 12); and (6) infarcted animals fed standard chow with 5% açai pulp (IA5, n = 12). Supplementation was performed for 90 days. After this period, the animals were evaluated using the final echocardiographic exam.

### Experimental MI

Animals were anesthetized with ketamine (70 mg/kg) and xylazine (1 mg/kg); next, the heart was exteriorized, and the left coronary artery was ligated with 5–0 mononylon between the pulmonary outflow tract and left atrium. Heart was replaced in the thorax, the lungs were inflated by positive pressure, and the thoracotomy was closed [2, 30]. In the sham group, the same MI procedure was performed but without coronary occlusion.

### Açai supplementation

Açai pulp was purchased commercially (Icefruit®) from the same batch, homogenized, packed in one-liter bottles, and stored at -80˚C for later use in chow. The açai pulp was analyzed, and the total phenolic compounds expressed as a quantity of gallic acid were 170 mg/100 g [31], antioxidant activity was 48.3 g of 2,2-diphenyl-1-picrylhydrazyl (DPPH)/kg [32], and total anthocyanin was 15.6 mg/100g [33]. We also determined fat content (6%) and water content (88%). Supplementation doses of 2% and 5% were chosen based on De Souza et al. (2012) [34] and Fragoso et al. (2013) [35], respectively. Nuvilab chow (Nuvital®) was used for all animals. Chow was initially chopped for the later addition of açai pulp, and the mixture was pelleted and dried at ambient temperature. The chow was stored at -20˚C.

Seven days after surgery, an initial echocardiographic exam was performed, and the animals were allocated into groups to start açai supplementation for 3 months. The food intake of all animals was measured periodically. The mean daily intake of each rat was then calculated.

## Echocardiographic study

At both time points (i.e., initial echocardiographic exam 7 days after surgery and final exam after 90 days of supplementation), animals were evaluated using a transthoracic echocardiographic exam. All measurements were performed according to the method recommended by the American Society of Echocardiography [36] and validated in the infarcted rat model in our laboratory [37]. The equipment used was General Electric Medical Systems, model Vivid S6 (Tirat Carmel, Israel) with a multifrequency transducer of 5 to 11.5 MHz. Mitral inflow and aortic flow were assessed using the same transducer operating at 5.0 MHz.

The initial echocardiographic examination was performed on infarcted animals to check the infarct size and fraction area variation, to homogenize infarct size among the groups. In the final echocardiographic examination, cardiac structures were measured in at least five consecutive cardiac cycles [38, 39].

## Morphometric analysis

Transverse sections (5 μm) of the left ventricles (LVs) were performed in paraffin blocks and stained with hematoxylin and eosin for cardiomyocyte cross-sectional area (CSA), and with Picrosirius Red Stain for interstitial collagen fraction (ICF) and infarction size calculations. The infarct size was calculated using the following formula: [(endocardial + epicardial circumferences of the infarcted area) / (endocardial + epicardial ventricular circumferences) × 100] (**Fig 1**). This method is considered more precise than echocardiography analysis because echocardiogram measure only the endocardial perimeter of the infarct, which may be overestimated by the infarction expansion process [40]. In this study, animals with infarcted area smaller than 30% were excluded.

CSA was measured from 70 cells per ventricle. Digital images were collected (400 × magnification) using a video camera attached to a Leica microscope. ICF was determined in remote cardiac areas free from MI from at least 30 digital images (400 × magnification). All images were collected with a video camera attached to a Leica microscope, and the images were analyzed with the Image-Pro Plus 3.0 software program (Media Cybernetics, Silver Spring, MD, USA).

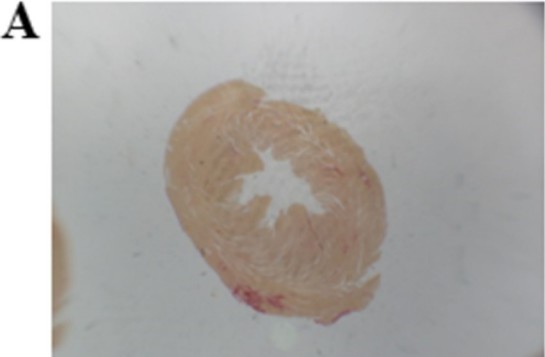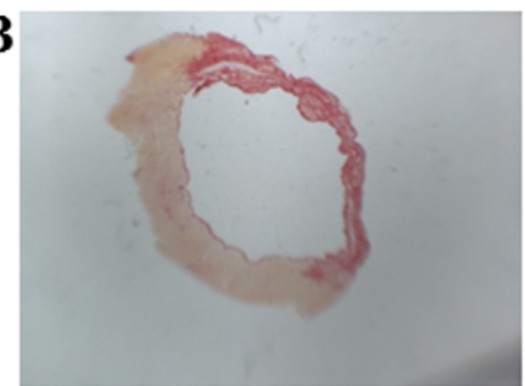

**Fig 1. Myocardial infarcted area.** (A) Sham animal heart, (B) Infarcted animal heart.

## Lipid hydroperoxide, oxidative stress, and energy metabolism enzymes

Protein extraction was performed on LV samples to determine the lipid hydroperoxide (LH) concentration and enzyme activities. Glutathione peroxidase (GPx, E.C.1.11.1.9), superoxide dismutase (SOD, E.C.1.15.1.1), and catalase (CAT, E.C.1.11.1.6) activities were assessed as previously described [41, 42]. Cardiac energy metabolism was assessed using $\beta$-hydroxyacyl coenzyme-A dehydrogenase ($\beta$-OHADH, E.C.1.1.1.35.), lactate dehydrogenase (LDH, E.C.1.1.1.27), pyruvate dehydrogenase complex (PDH), citrate synthase (CS; E.C.4.1.3.7.), complex I (NADH: ubiquinone oxidoreductase), complex II (succinate dehydrogenase), and ATP synthase (EC 3.6.3.14) activities, as previously described [42].

Spectrophotometric determinations were performed using a Pharmacia Biotech Spectrophotometer UV/visible Ultrospec 5000 with Swift II Application software for computer system control, 974213 (Cambridge, England, UK) at 560 nm.

## Malondialdehyde concentration

Protein extraction was performed on LV samples and quantified using the Bradford method [43]. The extraction was conducted in red light 1 day before the analysis. MDA concentration was determined by the method in (Nielsen et al. [1997]). Fluorometric detection was performed at 527 nm of excitation and 551 nm of emission [44].

## Cytokine production and tissue inhibitor of metalloproteinase-1 evaluation

Tumor necrosis factor-$\alpha$ (TNF-$\alpha$), IFN-$\gamma$, interleukin-10 (IL-10) and tissue inhibitor of metalloproteinase-1 (TIMP-1) concentrations were determined. Protein extraction was performed on the LV samples and quantified using the Bradford method [43]. The extraction was evaluated by ELISA, according to the manufacturer's instructions (R&D Systems, Minneapolis, MN, USA).

## Western blot analysis

Nuclear erythroid factor-2 (Nrf-2), NF-κB, and pNF-κB total and phosphorylated (NF-κB and pNF-κB, respectively), collagen I, collagen III, and caspase-3 expression were determined. LV samples were extracted using radioimmunoprecipitation assay buffer or nuclear extraction buffer. Protein content was quantified using the Bradford method [43], and samples containing 50 µg of protein were separated by electrophoresis using a Mini-Protean 3 Electrophoresis Cell (Bio-Rad, Hercules, CA, USA) system and transferred to a nitrocellulose membrane. The membrane was blocked and incubated with primary antibodies. The membrane was then washed with TBS and Tween 20 and incubated with the appropriate secondary peroxidase-conjugated antibody. A SuperSignal® West Pico Chemiluminescent Substrate (Pierce Protein Research Products, Rockford, IL, USA) was used to detect bound antibodies. Glyceraldehyde-3-phosphate dehydrogenase (GAPDH) mouse monoclonal (Santa Cruz Biotechnology Inc., Europe) was used for normalization.

## Statistical analysis

Data are presented as the mean ± standard error of the mean (SEM). Variables with no normal distribution were normalized using the most suitable mathematical transformation. For data that could not be normalized, we used the Kruskal Wallis test (comparison among sham groups SA0, SA2, and SA5 and among infarcted groups IA0, IA2, and IA5) and the Mann Whitney test (comparison between groups: SA0 and IA0; SA2 and IA2; and SA5 and IA5). Then, Bonferroni correction was performed.

Data were analyzed by 2-factor ANOVA: (1) factor one: presence of MI (I); (2) factor two: açaí supplementation (A); and (3) interaction between factors I and A. When a significant interaction between factors was observed, Holm Sidak multiple comparisons were used. When no interaction was found, the factors were separated to assess whether MI animals differed from sham animals and whether supplemented animals differed from those without supplementation.

The results are as follows: (1) changes promoted by MI, corresponding to comparisons between the SA0 and IA0 groups in case of interactions, and comparisons between all animals with or without MI in the absence of interaction between the factors; (2) the effect of açaí supplementation in MI animals, corresponding to the comparison among groups IA0, IA2, and IA5 in the presence of interaction; and (3) the effect of açaí supplementation in all animals in the absence of interaction between the factors. Thus, in this last situation, we interpreted the results by indicating the effect of açaí supplementation in a general action in the heart. In this manner, we also observed this action in MI.

One-factor ANOVA was used to compare infarction size in the infarcted groups. Spearman's rank correlation coefficient test was performed to compare the dose-dependent effects of açaí supplementation. Differences were considered statistically significant if $P<0.05$. Statistical analyses were performed using SigmaStat for Windows software version 3.5 (Systat Software Inc. San Jose, CA, USA).

## Results

### Survival, food intake, and body weight

Mortality during the experimental period was as follows: one animal of the sham animals, eight animals from IA0, six from IA2, and five from IA5 ($P = 0.728$). As expected, no differences were observed in the initial echocardiographic examination (Table 1). In addition, no difference was observed in the infarction size among the infarcted groups in relation to food intake and body weight (Table 2).

### Effect of MI in the rat heart

MI promoted myocardial remodeling by morphological changes (higher LV systolic and diastolic diameter adjusted by body weight, higher LV mass index, left atrium, and CSA) and functional changes (systolic: lower fractional area change and ejection fraction; and diastolic: lower E' media and A' media) (Table 2).

Changes in energy metabolism were as follows: abnormal oxidation of fatty acids (lower β-OHADH), lower glucose oxidation (lower activity of PDH), lower CS, lower electron transport chain complexes (lower activity of complex I and complex II), lower ATP synthase, and higher

**Table 1. Initial echocardiographic exam (7 days after surgery).**

|  | Groups | | | P value |
|---|---|---|---|---|
|  | IA0 (n = 11) | IA2 (n = 12) | IA5 (n = 12) | P |
| Diastolic area (mm$^2$) | 65.3±3.32 | 66±.2.31 | 60.3±1.66 | 0.209 |
| Sistolic area (mm$^2$) | 44.4±2.74 | 44.8±2.85 | 40.1±2.14 | 0.362 |
| % MI | 38±2.96 | 37.3±2.94 | 36.2±2.38 | 0.899 |
| FAC (%) | 32.3±1.99 | 32.6±2.57 | 33.8±2.39 | 0.890 |

I: infarction; S: sham; A: açaí; A0: no supplementation; A2: 2% of açaí supplementation; A5: 5% of açaí supplementation. FAC: fractional area change. Data are expressed as mean ± SEM. it was performed 1-factor ANOVA.

**Table 2. Echocardiographic and morphometric data.**

| | Groups | | | | | | P values | | |
|---|---|---|---|---|---|---|---|---|---|
| | SA0 (n = 14) | SA2 (n = 13) | SA5 (n = 14) | IA0 (n = 12) | IA2 (n = 12) | IA5 (n = 12) | P (I) | P (A) | P (IxA) |
| Food intake (g) | 23±0.6 | 22.6±0.7 | 23.7±0.6 | 24.8±0.5 | 23.1±0.5 | 23.2±0.8 | 0.257 | 0.254 | 0.194 |
| Initial BW (g) | 266±9 | 260±9 | 275±8 | 252±5 | 260±8 | 258±7 | 0.133 | 0.605 | 0.595 |
| Final BW (g) | 430±10 | 432±14 | 436±11 | 452±11 | 445±13 | 447±13 | 0.127 | 0.949 | 0.906 |
| Infarction size (%) | - | - | - | 42.8±1.9 | 39.7±2.7 | 42.1±1.8 | - | 0.556 | - |
| *LVSD/BW (mm/kg) | 9.52±0.43 | 9.39±0.29 | 8.81±0.33 | 17.5±0.98 | 17.6±1.09 | 18.5±0.84 | <0.001[1] | 0.994 | 0.297 |
| *LVDD/BW (mm/kg) | 18.4±0.3 | 18.8±0.5 | 17.8±0.5 | 22.7±0.8 | 22.9±1 | 23.3±0.8 | <0.001[1] | 0.901 | 0.428 |
| *LVMI (g/kg) | 1.7±0.04 | 1.8±0.07 | 1.7±0.07 | 3.3±0.21 | 3±0.2 | 3.2±0.22 | <0.001[1] | 0.921 | 0.217 |
| #LA/BW (mm/kg) | 12±0.3 | 13±0.4 | 12±0.3 | 16±1 | 15±1.1 | 15±0.9 | 0.005[1] | 0.132 | 0.665 |
| #FAC (%) | 69±1.7 | 67±1.1 | 68±1.2 | 32±3 | 33±1.8 | 31±3.3 | <0.005[1] | 0.538 | 0.605 |
| *Ejection fraction | 86±1.3 | 87±1.2 | 88±0.7 | 54±3.8 | 55±3.5 | 50±3.2 | <0.001[1] | 0.794 | 0.338 |
| E' media (cm/s) | 3.9±0.2 | 4.1±0.2 | 4.1±0.1 | 3.9±0.2 | 3.6±0.2 | 3.6±0.1 | 0.012[1] | 0.988 | 0.176 |
| *A' media (mm²) | 3.7±0.2 | 3.5±0.2 | 3.3±0.2 | 2.9±0.2 | 2.9±0.3 | 3.5±0.4 | 0.013[1] | 0.583 | 0.100 |
| CSA (µm²) | 207±34 | 252±27 | 245±25 | 304±28 | 275±30 | 289±28 | 0.027[1] | 0.923 | 0.422 |

I: infarction; S: sham; A: açai; A0: no supplementation; A2: 2% açai supplementation; A5: 5% açai supplementation; BW: body weight; LVSD: left ventricular systolic diameter; LVDD: left ventricular diastolic diameter; LVMI: left ventricular mass index; LA: left atrium; FAC: fractional area change; E' media: average between early diastolic wave of the mitral annulus lateral and septal; A' media: average between late diastolic wave of the mitral annulus lateral and septal; CSA: cardiomyocyte cross-sectional area. Data are expressed as mean ± SEM. Bold numbers represents significant effects considered.

[1] Comparisons for I factor: infarcted animals different from sham animals.

* variables normalized for 2-factor ANOVA test.

# Mann Whitney and Kruskal Wallis.

lactate production from pyruvate (higher LDH activity). No difference was observed in phosphofructokinase (PFK) activity (Table 3). An antioxidant imbalance was also observed (higher LH and MDA concentrations, higher SOD activity, lower GPx activity, and Nrf-2 expression).

**Table 3. Energy metabolism markers.**

| | Groups | | | | | | P values | | |
|---|---|---|---|---|---|---|---|---|---|
| | SA0 (n = 8) | SA2 (n = 8) | SA5 (n = 8) | IA0 (n = 8) | IA2 (n = 8) | IA5 (n = 8) | P (I) | P (A) | P (IxA) |
| β-OHADH (nmol/g) | 33.1±2.02 | 33.6±1.87 | 37.2±1.54 | 20.2±1.45 | 19.0±1.6 | 24.6±1.41 | <0.001[1] | 0.014[2] | 0.823 |
| #LDH (nmol/g) | 70.3±2.66[a] | 81.1±7.27 | 65.6±6.42 | 139±6.39[a,A,C] | 70.1±1.77[A] | 65.4±2.16[C] | <0.005 | 0.880 | <0.005[3] |
| #PFK (nmol/g) | 143±5.9 | 141±9.2 | 133±9.1 | 155±4.67 | 152±5.4 | 143±9.2 | 0.129 | 0.939 | 0.447 |
| PDH (nmol/g) | 286±12.1 | 300±15.5 | 317±15.5 | 218±15.4 | 293±15.6 | 298±11.4 | 0.011[1] | <0.001[2] | 0.085 |
| *CS (nmol/g) | 86.3±8.15 | 108±10.2 | 131±9.87 | 51.8±2.37 | 62.6±5.27 | 62.6±3.95 | <0.001[1] | 0.002[2] | 0.291 |
| *complex I (nmol/g) | 5.19±0.07[a] | 4.99±0.14[b] | 5.11±0.10 | 3.39±0.29[a,C] | 3.37±0.35[b,B] | 4.78±0.22[B,C] | <0.001 | 0.015 | 0.015[3] |
| complex II (nmol/g) | 5.54±0.34 | 4.66±0.34 | 4.73±0.31 | 3.09±0.16 | 2.99±0.18 | 3.48±0.25 | <0.001[1] | 0.208 | 0.094 |
| ATP synthase (nmol/g) | 37.4±0.91[a, E] | 35.9±1.16[b,D] | 31.4±1.39[c,D,E] | 20.6±1.01[a] | 21.9±1.06[b] | 20.1±0.98[c] | <0.001 | 0.007 | 0.049 |

I: infarction; S: sham; A: açai; A0: no supplementation; A2: 2% açai supplementation; A5: 5% açai supplementation. β-OHADH: β-hydroxyacyl coenzyme-A dehydrogenase; LDH: lactate dehydrogenase; PFK: Phosphofructokinase; PDH: pyruvate dehydrogenase complex; CS: citrate synthase. Data are expressed as mean ± SEM. Bold numbers represents significant effects considered.

[1] Comparisons for I factor: infarcted animals different from sham animals.

[2] Comparisons for açai factor, β-OHADH: animals A0≠animals A5 and animals A2≠animals A5; PDH: animals A0≠animals A2 and animals A0≠animals A5; CS: animals A0≠animals A2 and animals A0≠animals A5.

[3] IxA: when interactions are observed, same superscript letters represents differences (P<0.05).

* variables normalized for 2-factor ANOVA test.

# Mann Whitney and Kruskal Wallis.

**Table 4. Oxidative stress markers.**

| | Groups | | | | | | P values | | |
|---|---|---|---|---|---|---|---|---|---|
| | SA0 (n = 8) | SA2 (n = 8) | SA5 (n = 8) | IA0 (n = 8) | IA2 (n = 8) | IA5 (n = 8) | P (I) | P (A) | P (IxA) |
| *LH (nmol/mg) | 250±10.9 | 241±13.5 | 251±11.9 | 301±15 | 286±15.2 | 254±15.4 | 0.004[1] | 0.251 | 0.169 |
| MDA (µmol/g) | 0.35±0.05[a] | 0.36±0.03[b] | 0.31±0.04 | 1.32±0.25[a,A,C] | 0.77±0.12[b,A] | 0.61±0.12[C] | <0.001 | 0.013 | 0.023[3] |
| CAT (µmol/mg) | 64.5±4.23 | 59.1±5.11 | 56.6±4.04 | 47.8±3.17 | 61±3.62 | 60.5±7 | 0.351 | 0.709 | 0.065 |
| SOD (nmol/mg) | 7.06±0.36[a] | 7.34±0.36[b] | 7.59±0.31 | 8.95±0.38[a,C] | 8.74±0.54[b,B] | 6.9±0.41[B,C] | 0.011 | 0.095 | 0.006[3] |
| GPx (nmol/mg) | 45.0±3.1 | 43.9±1.87 | 45.6±1.82 | 34.6±1.44 | 39.9±2.31 | 43.6±2.92 | 0.006[1] | 0.133 | 0.179 |
| (n) Espression Nrf-2 | (8) 1.45±0.31 | (8) 1.48±0.39 | (7) 1.21±0.38 | (6) 1.05±0.34 | (8) 0.66±0.18 | (8) 0.75±0.16 | 0.029[1] | 0.674 | 0.760 |

I: infarction; S: sham; A: açai; A0: no supplementation; A2: 2% açai supplementation; A5: 5% açai supplementation. LH: lipid hydroperoxide; MDA: malondialdehyde; CAT: catalase; SOD: superoxide dismutase; GPx: glutathione peroxidase; Nrf-2: expression of nuclear factor erythroid-2. Data are expressed as mean ± SEM. Bold numbers represents significant effects considered.

[1] Comparisons for I factor: infarcted animals different from sham animals.

[3] IxA: when interactions are observed, same superscript letters represents differences (P<0.05).

*variables normalized for 2-factor ANOVA test.

No difference was observed in CAT activity (**Table 4**). MI led to inflammatory alterations (higher IL-10 and lower INF-γ concentrations). No differences were observed in NF-κB total and phosphorylated (NF-κB and pNF-κB, respectively) expression (**Table 5 and S1 Fig**). A higher deposition of collagen was observed after MI (higher concentration of TIMP-1, percentage of ICF, and expression of collagen I). No differences were observed in the expression of collagen III and caspase-3 (**Table 6 and S2 and S3 Figs**).

## Effect of açai supplementation after MI

Açai supplementation in infarcted animals improved cardiac energy metabolism (higher activity of $\beta$-OHADH, PDH, CS, complex I, and lower LDH activity) (**Table 3 and Fig 2**) and attenuated oxidative stress (lower concentration of MDA and SOD activity) (**Table 4 and Fig 3**). Furthermore, açai supplementation modulated the inflammatory process (lower concentration of IL-10) (**Table 5 and Fig 4**) and decreased the deposition of collagen (lower concentration of TIMP-1 and percentage of ICF) (**Table 6 and Fig 4**).

## Dose-dependent effects of açai supplementation after MI

Different doses of açai supplementation led to dose-dependent effects. This effect can be observed in antioxidant enzymes: lower SOD activity ($P = 0.005$), and higher GPx activity ($P = 0.014$); lipid peroxidation markers: lower concentration of LH ($P = 0.034$), and MDA ($P = 0.009$); in energy metabolism enzymes: higher activity of PDH ($P = 0.001$), CS ($P = 0.043$), and complex I ($P = 0.003$); and lower activity of LDH ($P<0.001$); anti-inflammatory cytokine: lower concentration of IL-10 ($P<0.001$); and collagen deposit markers: lower percentage of ICF ($P = 0.016$), and lower concentration of TIMP-1 ($P<0.001$) (**S1 Table**).

## Discussion

The aim of this study was to analyze the influence of açai supplementation on cardiac remodeling after MI in rats. Our data showed that MI promoted morphological and functional cardiac alterations, altered energy metabolism, increased oxidative stress, worsened the inflammatory process, and increased collagen deposition in the heart. These alterations are characteristic of cardiac remodeling and have been observed in other studies [7–9]. Supplementation of açai in

**Table 5. Production of cytokines inflammatory.**

| | Groups | | | | | | P values | | |
|---|---|---|---|---|---|---|---|---|---|
| | SA0 | SA2 | SA5 | IA0 | IA2 | IA5 | P (I) | P (A) | P (IxA) |
| (n) IL-10 (pg/mg) | (6) 3.21±0.47[a,E] | (6) 4.81±0.71[D] | (6) 8.49±1.[c,D,E] | (6) 6.83±0.44[a,A,C] | (6) 4.75±0.62[A,B] | (6) 2.48±0.27[c,B,C] | 0.123 | 0.529 | **<0.001**[3] |
| INF-γ (pg/mg) | (6) 0.48±0.19 | (5) 1.05±0.32 | (5) 1.33±0.22 | (6) 0.48±0.18 | (6) 0.62±0.27 | (4) 0.13±0.08 | **0.008**[1] | 0.276 | 0.054 |
| [#]TNF-α (pg/mg) | (6) 0.10±0.01 | (6) 0.02±0.02 | (6) 0.78±0.47 | (6) 0.24±0.16 | (6) 0.92±0.49 | (6) 0.53±00.33 | 0.699 | 0.250 | 0.427 |
| Expression p NF-κB | (8) 0.85±0.10 | (8) 1.24±0.14 | (6) 1.08±0.21 | (6) 1.25±0.21 | (6) 1.03±0.27 | (7) 1.06±0.16 | 0.696 | 0.872 | 0.222 |
| Expression NF-κB | (8) 0.94±0.16 | (8) 1.16±0.12 | (6) 1.09±0.25 | (6) 1.20±0.17 | (6) 0.96±0.20 | (7) 1.03±0.17 | 0.990 | 0.997 | 0.415 |
| [*]Expression p NF-κB / NF-κB | (8) 1.00±0.10 | (8) 1.05±0.05 | (6) 0.97±0.13 | (6) 1.02±0.07 | (6) 0.86±0.06 | (7) 0.91±0.11 | 0.191 | 0.663 | 0.377 |

I: infarction; S: sham; A: açai; A0: no supplementation; A2: 2% açai supplementation; A5: 5% açai supplementation. IL-10: interleukin 10; INF-γ: interferon γ; TNF-α: tumor necrosis factor-α, NF-κB: nuclear factor kappa B. Data are expressed as mean ± SEM. Bold numbers represents significant effects considered.

[1] Comparisons for I factor: infarcted animals different from sham animals.

[3] IxA: when interactions are observed, same superscript letters represents differences (P<0.05).

[*]variables normalized for 2-factor ANOVA test.

[#]Mann Whitney and Kruskal Wallis.

infarcted animals improved energy metabolism, attenuated oxidative stress, modulated the inflammatory process, and decreased fibrosis. Furthermore, açai supplementation resulted in a dose-dependent response.

## Influence of açai pulp supplementation in MI heart

Under physiological conditions, fatty acids are used as the main energy substrate. However, after MI, cardiac metabolism changes and glucose oxidation increase energy production [45]. Açai supplementation increased fatty acid oxidation and decreased the glycolytic pathway after MI. These data show that açai increased the use of fatty acids as an energy substrate, which is similar to what occurs in the heart under normal conditions [45]. Moreover, as afore mentioned, açai pulp has a high lipid content (48% of lipids) [13, 14], which may have contributed to the increased use of fatty acids. Furthermore, açai supplementation improved the citric acid cycle and electron transport chain. In the citric acid cycle, acetyl coenzyme A (acetyl CoA) and oxaloacetate are catalyzed by citrate synthase to citrate. At the end of this cycle, it generates additional coenzymes,

**Table 6. Collagen degradation markers and caspase-3.**

| | Groups | | | | | | P values | | |
|---|---|---|---|---|---|---|---|---|---|
| | SA0 | SA2 | SA5 | IA0 | IA2 | IA5 | P (I) | P (A) | P (IxA) |
| [*]TIMP-1 (pg/mg) | (6) 8.94±2.48[a] | (4) 9.42±0.90 | (2) 18.1±8.85 | (4) 51.4±13.5[a,A,C] | (4) 12.3±1.62[A] | (4) 8.25±1.95[C] | 0.055 | **0.049** | **<0.001**[3] |
| [*]ICF (%) | (8) 2.19±0.3[a,E,F] | (8) 3.09±0.9[b,F] | (8) 3.08±0.3[c,E] | (8) 10.4±1.5[a,C] | (8) 10.2±1.4[b,B] | (8) 8.07±1.8[c,B,C] | **<0.001** | 0.090 | **<0.001**[3] |
| Expression Collagen I | (8) 0.73±0.13 | (6) 0.87±0.17 | (6) 0.73±0.16 | (6) 1.22±0.21 | (6) 1.18±0.29 | (6) 0.85±0.14 | **0.050**[1] | 0.439 | 0.603 |
| Expression Collagen III | (7) 0.84±0.09 | (6) 1.04±0.14 | (6) 1.26±0.24 | (6) 1.14±0.17 | (6) 1.00±0.07 | (6) 1.25±0.14 | 0.503 | 0.164 | 0.464 |
| Expression Caspase-3 | (8) 1.88±0.58 | (7) 2.81±0.73 | (8) 3.21±0.65 | (7) 1.58±0.42 | (8) 2.63±0.68 | (7) 2.36±0.65 | 0.397 | 0.182 | 0.853 |

I: infarction; S: sham; A: açai; A0: no supplementation; A2: 2% açai supplementation; A5: 5% açai supplementation. TIMP-1: tissue inhibitor of metalloproteinase-1; ICF: interstitial collagen fraction. Data are expressed as mean ± SEM. Bold numbers represents significant effects considered.

[1] Comparisons for I factor: infarcted animals different from sham animals.

[3] IxA: when interactions are observed, same superscript letters represents differences (P<0.05).

[*]variables normalized for 2-factor ANOVA test.

[#]Mann Whitney and Kruskal Wallis.

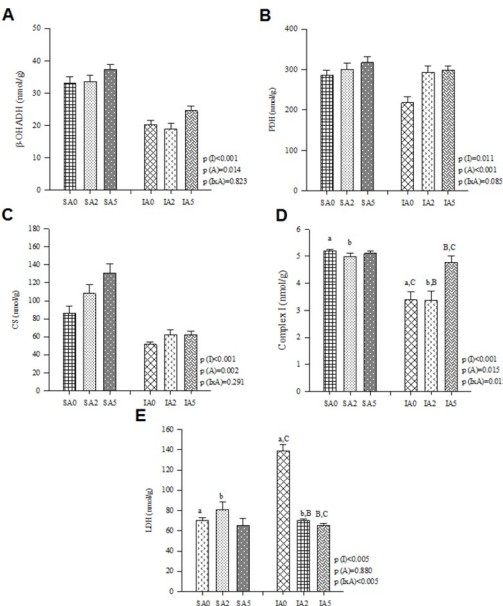

**Fig 2. Energy metabolism enzymes in sham and infarcted rats with and without açai supplementation.** (A) *β*-OHADH: *β*-hydroxyacyl coenzyme-A dehydrogenase, (B) PDH: pyruvate dehydrogenase complex, (C) CS: citrate synthase, (D) Complex I, (E) LDH: lactate dehydrogenase. Data are expressed as mean ± SEM. Same letter represents a significant difference between groups (p<0.005). Sample size: SA0 = 8; SA2 = 8; SA5 = 8; IA0 = 8; IA2 = 8; IA5 = 8.

which are responsible for the electron transport that occurs in the inner membrane of mitochondria to the intermembrane space. Moreover, this electron transfer is mediated by specific complexes I to IV and provides protons for complex V to produce ATP [45, 46]. Thus, açai supplementation improved energy metabolism in the heart. Two other studies have also observed that açai pulp supplementation improves cardiac energy metabolism in the heart in a doxorubicin-induced cardiotoxicity model [27] and in a global ischemia-reperfusion model [29].

After MI, there is an imbalance between RS production and myocardial antioxidant reserves [47]. In addition, changes in cardiac metabolism, increase the generation of superoxide radicals and increase oxidative stress [45]. In oxidative stress, the membrane is one of the cell components most affected by lipid peroxidation in its phospholipids, which causes changes in cell structure and permeability. This lipid peroxidation produces MDA, which is the most abundant end product of lipid peroxidation chain reactions and is commonly used as an indicator of oxidative damage [48, 49]. In our study, açai supplementation decreased the MDA concentration in the heart, leading to an improvement in oxidative stress. In addition, açai supplementation decreased the SOD activity. SOD is an antioxidant enzyme that protects mitochondria against deleterious superoxide radicals in pathophysiological and pathological conditions [50]. Other authors have also observed lower SOD activity in animals supplemented with açai. They suggested that the high levels of antioxidant compounds present in the açai pulp can reduce RS, maintaining redox balance, without the physiological need to increase antioxidant enzyme activity [51–53]. Therefore, the presence of phytochemical compounds, especially polyphenols, has been associated with the effects of açai as an antioxidant via both direct [13] and indirect effects [53–55] in different experimental models. Thus, in this study, the decreased MDA concentration and SOD activity promoted by açai supplementation suggest a reduction in RS in these animals, maintaining redox balance. Decreased SOD activity has also been reported with infarcted animals and supplementation of natural products with antioxidant/anti-inflammatory properties [7, 8].

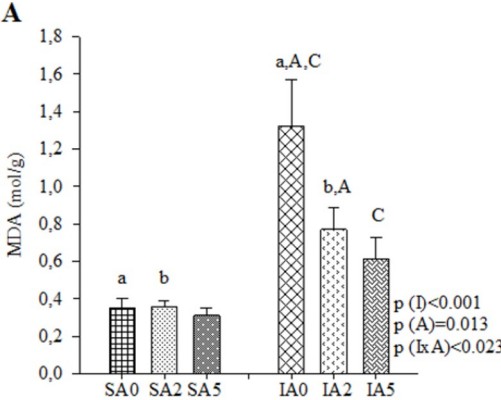

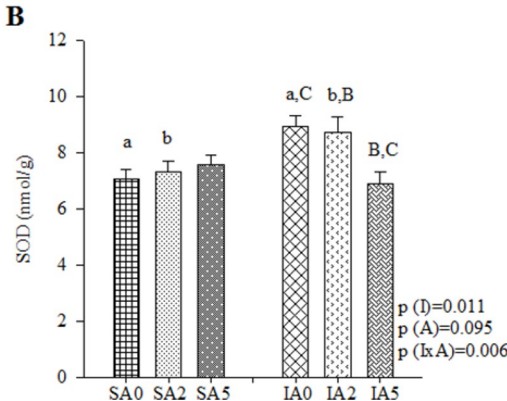

**Fig 3. Oxidative stress markers in sham and infarcted rats with and without açai supplementation.** (A) MDA: malondialdehyde, (B) SOD: superoxide dismutase. Data are expressed as mean ± SEM. The same letter represents a significant difference between groups (p<0.005). Sample size: SA0 = 8; SA2 = 8; SA5 = 8; IA0 = 8; IA2 = 8; IA5 = 8.

Regarding the inflammatory response, MI activates innate immunity, characterized by an increase in proinflammatory cytokines (TNF-$\alpha$, interleukin-1 $\beta$ (IL-1$\beta$), and interleukin-6 (IL-6)) [56]. An anti-inflammatory repair response is initiated, characterized by the activation and accelerated proliferation of fibroblasts [57, 58]. Finally, repairing macrophages are recruited and release inhibitory mediators (i.e., transforming growth factor-$\beta$ and IL-10), which suppress inflammation and activate profibrotic processes [59]. Thus, the acute inflammatory process after MI is followed by inflammatory balance and tissue repair with stabilization of the healing process [60]. Therefore, the decreased concentration of the anti-inflammatory cytokine IL-10, observed in groups supplemented with açai, suggests that the inflammatory process is balanced and that there is no reason to increase the production of anti-inflammatory cytokines. This balance in the inflammatory process can also be confirmed by the normal values of TNF-$\alpha$ and NF-$\kappa$B (phosphorylated, total, and its relation).

The participation of the extracellular matrix and its structural elements, such as collagen and other proteins, is also relevant in the remodeling process. The ECM is upregulated by matrix metalloproteinases (MMPs) and downregulated by tissue inhibitors of metalloproteinases (TIMPs) [61]. A study reported that MI promoted an increase in MMP-2 [8]. Therefore, an increase in TIMP-1 in infarcted animals is expected to be a consequence of the increase in MMPs [61]. In our study, açai supplementation decreased TIMP-1 concentration and ICF

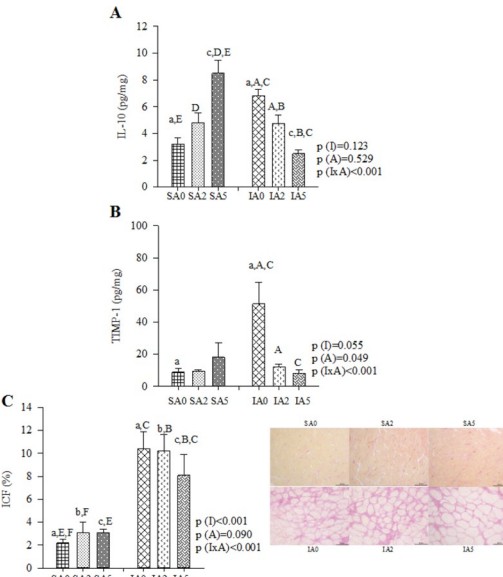

**Fig 4. Inflammatory process and collagen deposit in sham and infarcted rats with and without açai supplementation.** (A) IL-10: interleukin 10. Sample size: SA0 = 6; SA2 = 6; SA5 = 6; IA0 = 6; IA2 = 6; IA5 = 6. (B) TIMP-1: tissue inhibitor of metalloproteinase-1. Sample size: SA0 = 6; SA2 = 4; SA5 = 2; IA0 = 4; IA2 = 4; IA5 = 4. (C) ICF: interstitial collagen fraction. Sample size: SA0 = 8; SA2 = 8; SA5 = 8; IA0 = 8; IA2 = 8; IA5 = 8. Data are expressed as mean ± SEM. The same letter represents a significant difference between groups (p<0.005).

percentage. Thus, açai supplementation led to a decrease in extracellular matrix degradation and, consequently, a decrease in fibrosis.

This study did not observe an improvement in morphological and functional variables, as evaluated by echocardiography, promoted by açai supplementation. Our study supplemented açai for 3 months after MI, as prior studies performed in our laboratory with an experimental model of infarction have done [7, 8]. Considering these results with açai supplementation, we observed important biochemical improvements. We suggest that an increased supplementation time, would result in improvements in morphological and functional variables. In addition, açai supplementation doses had dose-dependent effects. Thus, increasing the dose of açai supplementation, would improvement the morphological and functional variables.

In this study, açai supplementation with 2% and 5% had a dose-dependent effect on post-MI cardiac remodeling. Regarding the açai dose, concentrations of 2% and 5% in rat chow were equivalent to 15.6 g (1 tablespoons daily) and 39 g (2.5 tablespoons daily) for humans, respectively [62]. Therefore, the amount is not large, açai is a product that can be easily found year-round in Brazil, and the price is affordable.

## Conclusion

Açai supplementation attenuated cardiac remodeling after MI in rats. The mechanism involved reduced oxidative stress, improved energy metabolism, modulated the inflammatory process, and decreased fibrosis. Different doses of açai supplementation had dose-dependent effects on cardiac remodeling.

## Supporting information

**S1 Table. Dose-dependent effects of different doses of açai supplementation.**
(DOCX)

**S1 Fig. NF-κB total and phosphorylated in sham and infarcted rats with and without açai supplementation.** (A) NF-κB total: nuclear factor kappa B, (B) pNF-κB: nuclear factor kappa B: phosphorylated nuclear factor kappa B. Sample size: SA0 = 8; SA2 = 8; SA5 = 6; IA0 = 6; IA2 = 6; IA5 = 7.
(TIF)

**S2 Fig. Collagen I and III in sham and infarcted rats with and without açai supplementation.** (A) Collagen I. Sample size: SA0 = 8; SA2 = 6; SA5 = 6; IA0 = 6; IA2 = 6; IA5 = 6. (B) Collagen III. Sample size: SA0 = 7; SA2 = 6; SA5 = 6; IA0 = 6; IA2 = 6; IA5 = 6.
(TIF)

**S3 Fig. Nrf-2 and caspase-3 in sham and infarcted rats with and without açai supplementation.** (A) Nrf-2: expression of nuclear factor erythroid-2. Sample size: SA0 = 8; SA2 = 8; SA5 = 7; IA0 = 6; IA2 = 8; IA5 = 8. (B) Caspase-3. Sample size: SA0 = 8; SA2 = 7; SA5 = 8; IA0 = 7; IA2 = 8; IA5 = 7.
(TIF)

## Acknowledgments

We thank Mario B. Bruno, José Georgette and Antonio Carlos de Lalla for animal management and technical assistance. AMF, SARP, BFP, MFM, LAMZ and PSAG designed the research; AMF, ACC, BPMR, RACS, AFG, TFB, LLI, BCO, BLBP, AAHF and KO conducted the research; AMF and SARP analyzed the data and wrote the paper. SARP had primary responsibility for final content. All authors read and approved the final manuscript.

## Author Contributions

**Conceptualization:** Paula Schmidt Azevedo, Leonardo Antonio Mamede Zornoff, Marcos Ferreira Minicucci, Bertha Furlan Polegato, Sergio Alberto Rupp Paiva.

**Data curation:** Amanda Menezes Figueiredo, Ana Carolina Cardoso, Renata Aparecida Candido Silva, Andrea Freitas Goncalves Della Ripa, Tatiana Fernanda Bachiega Pinelli, Bruna Camargo Oliveira, Bruna Paola Murino Rafacho, Larissa Lumi Watanabe Ishikawa, Paula Schmidt Azevedo, Katashi Okoshi, Ana Angelica Henrique Fernandes, Leonardo Antonio Mamede Zornoff, Marcos Ferreira Minicucci, Bertha Furlan Polegato, Sergio Alberto Rupp Paiva.

**Formal analysis:** Amanda Menezes Figueiredo, Bruna Paola Murino Rafacho, Paula Schmidt Azevedo, Katashi Okoshi, Ana Angelica Henrique Fernandes, Leonardo Antonio Mamede Zornoff, Marcos Ferreira Minicucci, Bertha Furlan Polegato, Sergio Alberto Rupp Paiva.

**Funding acquisition:** Sergio Alberto Rupp Paiva.

**Investigation:** Amanda Menezes Figueiredo, Bruna Leticia Buzati Pereira, Paula Schmidt Azevedo, Marcos Ferreira Minicucci, Bertha Furlan Polegato, Sergio Alberto Rupp Paiva.

**Methodology:** Amanda Menezes Figueiredo, Ana Carolina Cardoso, Paula Schmidt Azevedo, Leonardo Antonio Mamede Zornoff, Marcos Ferreira Minicucci, Bertha Furlan Polegato, Sergio Alberto Rupp Paiva.

**Project administration:** Amanda Menezes Figueiredo, Sergio Alberto Rupp Paiva.

**Resources:** Amanda Menezes Figueiredo, Sergio Alberto Rupp Paiva.

**Software:** Sergio Alberto Rupp Paiva.

**Supervision:** Leonardo Antonio Mamede Zornoff, Marcos Ferreira Minicucci, Sergio Alberto Rupp Paiva.

**Validation:** Amanda Menezes Figueiredo, Ana Carolina Cardoso, Bruna Leticia Buzati Pereira, Renata Aparecida Candido Silva, Andrea Freitas Goncalves Della Ripa, Tatiana Fernanda Bachiega Pinelli, Bruna Camargo Oliveira, Bruna Paola Murino Rafacho, Larissa Lumi Watanabe Ishikawa, Paula Schmidt Azevedo, Katashi Okoshi, Ana Angelica Henrique Fernandes, Leonardo Antonio Mamede Zornoff, Marcos Ferreira Minicucci, Bertha Furlan Polegato, Sergio Alberto Rupp Paiva.

**Visualization:** Amanda Menezes Figueiredo, Ana Carolina Cardoso, Bruna Leticia Buzati Pereira, Renata Aparecida Candido Silva, Andrea Freitas Goncalves Della Ripa, Tatiana Fernanda Bachiega Pinelli, Bruna Camargo Oliveira, Bruna Paola Murino Rafacho, Paula Schmidt Azevedo, Katashi Okoshi, Ana Angelica Henrique Fernandes, Leonardo Antonio Mamede Zornoff, Marcos Ferreira Minicucci, Bertha Furlan Polegato, Sergio Alberto Rupp Paiva.

**Writing – original draft:** Amanda Menezes Figueiredo, Paula Schmidt Azevedo, Katashi Okoshi, Leonardo Antonio Mamede Zornoff, Marcos Ferreira Minicucci, Bertha Furlan Polegato, Sergio Alberto Rupp Paiva.

**Writing – review & editing:** Amanda Menezes Figueiredo, Paula Schmidt Azevedo, Katashi Okoshi, Leonardo Antonio Mamede Zornoff, Marcos Ferreira Minicucci, Bertha Furlan Polegato, Sergio Alberto Rupp Paiva.

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
