## [Decision Letter · Decision Letter 0]

6 Dec 2021

PONE-D-21-31750Açai supplementation (Euterpe oleracea Mart.) attenuates cardiac remodeling after myocardial infarction in rats through different mechanistic pathwaysPLOS ONE

Dear Dr. Figueiredo,

Thank you for submitting your manuscript to PLOS ONE. After careful consideration, we feel that it has merit but does not fully meet PLOS ONE’s publication criteria as it currently stands. Therefore, we invite you to submit a revised version of the manuscript that addresses the points raised by the two reviewers (below) during the review process.

We look forward to receiving your revised manuscript.

Kind regards,

Luis Eduardo M Quintas, Ph.D.

Academic Editor

PLOS ONE

Journal Requirements:

2. As part of your revision, please complete and submit a copy of the Full ARRIVE 2.0 Guidelines checklist, a document that aims to improve experimental reporting and reproducibility of animal studies for purposes of post-publication data analysis and reproducibility: https://arriveguidelines.org/sites/arrive/files/Author%20Checklist%20-%20Full.pdf (PDF). Please include your completed checklist as a Supporting Information file. Note that if your paper is accepted for publication, this checklist will be published as part of your article.

 [This research was supported in part by Coordenação de Aperfeiçoamento Pessoal de Nível Superior (CAPES), Brazil. The funding source had no involvement in study design, in the collection, analysis and interpretation of data, in the writing of the report, and in the decision to submit the article for publication.]

4. PLOS requires an ORCID iD for the corresponding author in Editorial Manager on papers submitted after December 6th, 2016. Please ensure that you have an ORCID iD and that it is validated in Editorial Manager. To do this, go to ‘Update my Information’ (in the upper left-hand corner of the main menu), and click on the Fetch/Validate link next to the ORCID field. This will take you to the ORCID site and allow you to create a new iD or authenticate a pre-existing iD in Editorial Manager. Please see the following video for instructions on linking an ORCID iD to your Editorial Manager account: https://www.youtube.com/watch?v=_xcclfuvtxQ.

Reviewers' comments:

Reviewer's Responses to Questions

**Comments to the Author**

1. Is the manuscript technically sound, and do the data support the conclusions?

Reviewer #1: Yes

Reviewer #2: Partly

2. Has the statistical analysis been performed appropriately and rigorously? 

Reviewer #1: Yes

Reviewer #2: Yes

3. Have the authors made all data underlying the findings in their manuscript fully available?

Reviewer #1: No

Reviewer #2: No

4. Is the manuscript presented in an intelligible fashion and written in standard English?

Reviewer #1: Yes

Reviewer #2: Yes

5. Review Comments to the Author

Reviewer #1: The present study evaluated the beneficial effect of açaí pulp supplementation on cardiac remodeling after myocardial infarction in rats. The study showed evidence that açaí improves oxidative stress and energy metabolism, modulate inflammation, and decreases fibrosis. The article reports the results of a well-planned research study. However, some points have to be addressed:

Methods/abstract:

- Please, provide the protocol’s number of the local animal research ethics committee.

- Please, provide the origin of the açaí pulp. Did the authors use the same batch of açaí pulp during the whole experimental period?

- Please, clearly state (Açaí supplementation topic) when the animals started to receive the supplementation with açaí pulp and for how long. In the third paragraph of material and methods”, the authors describe that “After the initial echocardiographic exam, animals were allocated into six groups”. In the echocardiographic study, it is described that “the initial echocardiographic exam was 7 days after surgery”. Therefore, the initial supplementation with açaí was 7 days after surgery? This point is also not clear in the abstract.

Results:

- The authors did not describe the results of the following biomarkers:

PFK: Phosphofructokinase activity (table 3); CAT activity (table 4); NF-kB, pNF-kB, NF-kB / pNF-kB; Collagen III and Caspase3 (table 6)

- Are you sure that INF-y concentration is reduced in infarcted animals compared with sham?

- What is the explanation for the increased SOD activity in infarcted animals? The authors described that the açaí supplementation after MI attenuated the oxidative stress because of the lower concentration of MDA and SOD activity. Is the reduction of the antioxidant enzyme activity (SOD) indicative of reduced oxidative stress? The antioxidant effect of açaí is well known for increasing SOD activity in different tissues. Therefore, in this sudy, the attenuation of the oxidative stress by açaí supplementation seems to be due to a lower concentration of MDA and increased GPx activity.

Please, revise the text.

- Page 17 (lines 334-336): “Different doses of açai supplementation led to dose-dependent effects. This effect can be observed in oxidative stress enzymes: lower SOD activity (P=0.005) and higher activity of GPx (P=0.014).” In this paragraph, “the oxidative stress enzymes” should be replaced by “antioxidant enzymes.”

-The expression of the NF-kB, p NF-kB (table 5), collagen I and II, and caspa3 (table 6) determined by western blotting should be represented in graphics with the corresponding photos of the membranes.

- It would also be interesting to show the myocardial infarcted area and cardiac remodeling.

Discussion:

Lines 384-388: “The lower SOD activity has also been reported in other studies with infarcted animals and supplementation of natural products with antioxidant/anti-inflammatory properties, such as tomato and rosemary [7,8]. This antioxidant activity of açai was previously reported and was associated with the high concentration of anthocyanins and flavonoids present in this fruit [24,50].”

Are you sure that the reduced SOD activity induced by açaí supplementation may be considered an antioxidant activity? In this study, the GPX activity, an important antioxidant enzyme was reduced in MI animals and increased by açaí. The authors should discuss this result.

- The manuscript contains typographical English errors and must be carefully revised.

For example: (abstract, line 32) replace “n rats” with “in rats”…..

Reviewer #2: The article presents interesting data regarding the effects of açai on the infarcted heart and on the cardiac remodeling process. However, some questions need to be raised:

- The author does not present any results in the form of a graph, nor images of the histological and echocardiographic analyzes performed. You need to include these images as well as images from western blot analysis. In this way, readers were able to assess their quality and the ability to draw conclusions from them.

- Furthermore, how do the authors explain that the biochemical impacts of açai administration have not resulted in any morphological and functional improvement for the heart? It is necessary to discuss the validity of biochemical changes that do not result in functional or morphological improvements after the infarction or in the cardiac remodeling process

6. PLOS authors have the option to publish the peer review history of their article (what does this mean?). If published, this will include your full peer review and any attached files.

Reviewer #1: No

Reviewer #2: No

---

## [Author Response · Author response to Decision Letter 0]

21 Jan 2022

Dear Reviewers of Plos One, 

Thank you very much for your time, careful review and considerations. We tried to answer all your concerns. Also, we changed the manuscript including your valuable suggestions. 

Reviewer #1: 

Methods/abstract:

- Please, provide the protocol’s number of the local animal research ethics committee.

The study protocol (1066/2013-CEUA) was submitted and approved by the Botucatu Medical School Animal Research Ethics Committee. 

- Please, provide the origin of the açaí pulp. Did the authors use the same batch of açaí pulp during the whole experimental period?

Açai pulp was purchased commercially (Icefruit®) from the same batch, homogenized, packed in one-liter bottles, and stored at -80ºC for later use in chow.

- Please, clearly state (Açaí supplementation topic) when the animals started to receive the supplementation with açaí pulp and for how long. In the third paragraph of material and methods”, the authors describe that “After the initial echocardiographic exam, animals were allocated into six groups”. In the echocardiographic study, it is described that “the initial echocardiographic exam was 7 days after surgery”. Therefore, the initial supplementation with açaí was 7 days after surgery? This point is also not clear in the abstract.

Seven days after surgery, an initial echocardiographic exam was performed, and the animals were allocated into groups to start açai supplementation for 3 months. The food intake of all animals was measured periodically. The mean daily intake of each rat was then calculated. 

Results:

- The authors did not describe the results of the following biomarkers:

PFK: Phosphofructokinase activity (table 3); CAT activity (table 4); NF-kB, pNF-kB, NF-kB/pNF-kB; Collagen III and Caspase3 (table 6).

No difference was observed in phosphofructokinase (PFK) activity (Table 3). No difference was observed in CAT activity (Table 4). No differences were observed in NF-�B total and phosphorylated (NF-�B and pNF-�B, respectively) expression (Table 5, S1 Fig.). No differences were observed in the expression of collagen III and caspase-3 (Table 6, S2 Fig., and S3 Fig.).

- Are you sure that INF-y concentration is reduced in infarcted animals compared with sham?

Yes, the statistical test shows that INF-γ concentration is lower in infarcted animals (IA0, IA2 and IA5) compared with sham animals (SA0, SA2 and SA5), p(I)=0.008. The 2-factor ANOVA is comparing marginal data between infarcted animals and sham animals (infarcted animals: 0.45±0.13; sham animals: 0.92±0.16)

- What is the explanation for the increased SOD activity in infarcted animals?

Oxidative stress has been associated with diastolic dysfunction [1]. The toxic effects promoted by reactive species can be prevented, in part, by the antioxidant enzyme system, including SOD [2]. SOD is considered the first line of defense in protecting the mitochondria against deleterious effects of increased superoxide production, as described in cardiac remodeling and heart failure [3]. The increased SOD activity has also been reported in other studies with infarcted animals [4,5].

- The authors described that the açaí supplementation after MI attenuated the oxidative stress because of the lower concentration of MDA and SOD activity. Is the reduction of the antioxidant enzyme activity (SOD) indicative of reduced oxidative stress? 

The modification of SOD activity observed does not indicate attenuation of oxidative stress. However, cellular systems and enzymes, including the SOD, counterbalance the production of RS [6]. Thus, the decreased SOD activity in cardiac tissue of infarcted animals that received açai supplementation showed change in the pattern of RS production. Different from SOD, the MDA is the most abundant end product of lipid peroxidation chain reactions, which is commonly used as an indicator of oxidative damage [6,7].

- The antioxidant effect of açaí is well known for increasing SOD activity in different tissues. Therefore, in this sudy, the attenuation of the oxidative stress by açaí supplementation seems to be due to a lower concentration of MDA and increased GPx activity. Please, revise the text.

-Discussion: Lines 384-388: “The lower SOD activity has also been reported in other studies with infarcted animals and supplementation of natural products with antioxidant/anti-inflammatory properties, such as tomato and rosemary [7,8]. This antioxidant activity of açai was previously reported and was associated with the high concentration of anthocyanins and flavonoids present in this fruit [24,50].”

Are you sure that the reduced SOD activity induced by açaí supplementation may be considered an antioxidant activity? In this study, the GPX activity, an important antioxidant enzyme was reduced in MI animals and increased by açaí. The authors should discuss this result.

Other authors have also observed lower SOD activity in animals supplemented with açai. They suggested that the high levels of antioxidant compounds present in the açai pulp can reduce RS, maintaining redox balance, without the physiological need to increase antioxidant enzyme activity [8-10]. Therefore, the presence of phytochemical compounds, especially polyphenols, has been associated with the effects of açai as an antioxidant via both direct [11] and indirect effects [10,12,13] in different experimental models. Thus, in this study, the decreased MDA concentration and SOD activity promoted by açai supplementation suggest a reduction in RS in these animals, maintaining redox balance. 

 Regarding the GPx activity, the statistical test shows lower activity in infarcted animals (IA0, IA2 and IA5) compared with sham animals (SA0, SA2 and SA5), p(I)=0.006. The 2-factor ANOVA is comparing marginal data between infarcted animals and sham animals (infarcted animals: 39.4±1.49; sham animals: 44.9±1.30). No difference was observed in animals supplemented with açai pulp. 

- Page 17 (lines 334-336): “Different doses of açai supplementation led to dose-dependent effects. This effect can be observed in oxidative stress enzymes: lower SOD activity (P=0.005) and higher activity of GPx (P=0.014).” In this paragraph, “the oxidative stress enzymes” should be replaced by “antioxidant enzymes.”

“Different doses of açai supplementation led to dose-dependent effects. This effect can be observed in antioxidant enzymes: lower SOD activity (P=0.005) and higher activity of GPx (P=0.014).”

- It would also be interesting to show the myocardial infarcted area and cardiac remodeling.

 Fig. 1. Myocardial infarcted area.

(A) Sham animal heart, (B) Infarcted animal heart. 

Reviewer #1: The expression of the NF-kB, p NF-kB (table 5), collagen I and II, and caspa3 (table 6) determined by western blotting should be represented in graphics with the corresponding photos of the membranes.

Reviewer #2: The author does not present any results in the form of a graph, nor images of the histological and echocardiographic analyzes performed. You need to include these images as well as images from western blot analysis. In this way, readers were able to assess their quality and the ability to draw conclusions from them.

Fig. 2. Energy metabolism enzymes in Sham and infarcted rats with and without açai supplementation.

(A) β-OHADH: β-hydroxyacyl coenzyme-A dehydrogenase, (B) PDH: pyruvate dehydrogenase complex, (C) CS: citrate synthase, (D) Complex I, (E) LDH: lactate dehydrogenase. Data are expressed as mean ± SEM. Same letter represents a significant difference between groups (p<0.005). Sample size: SA0=8; SA2=8; SA5=8; IA0=8; IA2=8; IA5=8.

Fig. 3. Oxidative stress markers in Sham and infarcted rats with and without açai supplementation.

(A) MDA: malondialdehyde, (B) SOD: superoxide dismutase. Data are expressed as mean ± SEM. Same letter represents a significant difference between groups (p<0.005). Sample size: SA0=8; SA2=8; SA5=8; IA0=8; IA2=8; IA5=8.

Fig. 4. Inflammatory process and collagen deposit in Sham and infarcted rats with and without açai supplementation.

(A) IL-10: interleukin 10. Sample size: SA0=6; SA2=6; SA5=6; IA0=6; IA2=6; IA5=6. (B) TIMP-1: tissue inhibitor of metalloproteinase-1. Sample size: SA0=6; SA2=4; SA5=2; IA0=4; IA2=4; IA5=4. (C) ICF: interstitial collagen fraction. Sample size: SA0=8; SA2=8; SA5=8; IA0=8; IA2=8; IA5=8. Data are expressed as mean ± SEM. Same letter represents a significant difference between groups (p<0.005). 

S1 Fig. Expression of NF-�B total and phosphorylated in Sham and infarcted rats with and without açai supplementation.

(A) NF-�B total: nuclear factor kappa B, (B) pNF-�B: nuclear factor kappa B: phosphorylated nuclear factor kappa B. Sample size: SA0=8; SA2=8; SA5=6; IA0=6; IA2=6; IA5=7.

S2 Fig. Collagen I and III in Sham and infarcted rats with and without açai supplementation.

(A) Collagen I. Sample size: SA0=8; SA2=6; SA5=6; IA0=6; IA2=6; IA5=6. (B) Collagen III. Sample size: SA0=7; SA2=6; SA5=6; IA0=6; IA2=6; IA5=6.

S3 Fig. Nrf-2 and Caspase-3 in Sham and infarcted rats with and without açai supplementation.

(A) Nrf-2: expression of nuclear factor erythroid-2. Sample size: SA0=8; SA2=8; SA5=7; IA0=6; IA2=8; IA5=8. (B) Caspase-3. Sample size: SA0=8; SA2=7; SA5=8; IA0=7; IA2=8; IA5=7. 

- Furthermore, how do the authors explain that the biochemical impacts of açai administration have not resulted in any morphological and functional improvement for the heart? It is necessary to discuss the validity of biochemical changes that do not result in functional or morphological improvements after the infarction or in the cardiac remodeling process

This study did not observe an improvement in morphological and functional variables, evaluated by echocardiography, promoted by açai supplementation. Our study supplemented açai for 3 months after MI, as prior studies performed in our laboratory with an experimental model of infarction have done [4,5]. Considering these results with the açai supplementation, we observed important biochemical improvements. We suggest that an increased supplementation time, would result in improvements in morphological and functional variables. In addition, açai supplementation doses had dose-dependent effects. Thus, increasing the dose of açai supplementation, would improvement the morphological and functional variables.

References 

1. Jeong EM, Dudley SC, Jr. (2015) Diastolic dysfunction. Circ J 79: 470-477.

2. van Deel ED, Lu Z, Xu X, Zhu G, Hu X, et al. (2008) Extracellular superoxide dismutase protects the heart against oxidative stress and hypertrophy after myocardial infarction. Free Radic Biol Med 44: 1305-1313.

3. Liu T, Chen L, Kim E, Tran D, Phinney BS, et al. (2014) Mitochondrial proteome remodeling in ischemic heart failure. Life Sci 101: 27-36.

4. Murino Rafacho BP, Portugal Dos Santos P, Goncalves AF, Fernandes AAH, Okoshi K, et al. (2017) Rosemary supplementation (Rosmarinus oficinallis L.) attenuates cardiac remodeling after myocardial infarction in rats. PLoS One 12: e0177521.

5. Pereira BLB, Reis PP, Severino FE, Felix TF, Braz MG, et al. (2017) Tomato (Lycopersicon esculentum) or lycopene supplementation attenuates ventricular remodeling after myocardial infarction through different mechanistic pathways. J Nutr Biochem 46: 117-124.

6. Daiber A, Hahad O, Andreadou I, Steven S, Daub S, et al. (2021) Redox-related biomarkers in human cardiovascular disease - classical footprints and beyond. Redox Biol 42: 101875.

7. Li G, Ye Y, Kang J, Yao X, Zhang Y, et al. (2012) l-Theanine prevents alcoholic liver injury through enhancing the antioxidant capability of hepatocytes. Food Chem Toxicol 50: 363-372.

8. Barbosa PO, Souza MO, Silva MPS, Santos GT, Silva ME, et al. (2021) Acai (Euterpe oleracea Martius) supplementation improves oxidative stress biomarkers in liver tissue of dams fed a high-fat diet and increases antioxidant enzymes' gene expression in offspring. Biomed Pharmacother 139: 111627.

9. de Freitas Carvalho MM, Lage NN, de Souza Paulino AH, Pereira RR, de Almeida LT, et al. (2019) Effects of acai on oxidative stress, ER stress, and inflammation-related parameters in mice with high fat diet-fed induced NAFLD. Sci Rep 9: 8107.

10. de Souza MO, Silva M, Silva ME, Oliveira Rde P, Pedrosa ML (2010) Diet supplementation with acai (Euterpe oleracea Mart.) pulp improves biomarkers of oxidative stress and the serum lipid profile in rats. Nutrition 26: 804-810.

11. Schauss AG, Wu X, Prior RL, Ou B, Huang D, et al. (2006) Antioxidant capacity and other bioactivities of the freeze-dried Amazonian palm berry, Euterpe oleraceae mart. (acai). J Agric Food Chem 54: 8604-8610.

12. Guerra JF, Magalhaes CL, Costa DC, Silva ME, Pedrosa ML (2011) Dietary acai modulates ROS production by neutrophils and gene expression of liver antioxidant enzymes in rats. J Clin Biochem Nutr 49: 188-194.

13. Xie C, Kang J, Burris R, Ferguson ME, Schauss AG, et al. (2011) Acai juice attenuates atherosclerosis in ApoE deficient mice through antioxidant and anti-inflammatory activities. Atherosclerosis 216: 327-333.

---

## [Decision Letter · Decision Letter 1]

18 Feb 2022

Açai supplementation (Euterpe oleracea Mart.) attenuates cardiac remodeling after myocardial infarction in rats through different mechanistic pathways

PONE-D-21-31750R1

Dear Dr. Figueiredo,

We’re pleased to inform you that your manuscript has been judged scientifically suitable for publication and will be formally accepted for publication once it meets all outstanding technical requirements.

Kind regards,

Luis Eduardo M Quintas, Ph.D.

Academic Editor

PLOS ONE

Additional Editor Comments (optional):

Reviewers' comments:

Reviewer's Responses to Questions

**Comments to the Author**

1. If the authors have adequately addressed your comments raised in a previous round of review and you feel that this manuscript is now acceptable for publication, you may indicate that here to bypass the “Comments to the Author” section, enter your conflict of interest statement in the “Confidential to Editor” section, and submit your "Accept" recommendation.

Reviewer #2: All comments have been addressed

2. Is the manuscript technically sound, and do the data support the conclusions?

Reviewer #2: Yes

3. Has the statistical analysis been performed appropriately and rigorously? 

Reviewer #2: Yes

4. Have the authors made all data underlying the findings in their manuscript fully available?

Reviewer #2: Yes

5. Is the manuscript presented in an intelligible fashion and written in standard English?

Reviewer #2: Yes

6. Review Comments to the Author

Reviewer #2: (No Response)

7. PLOS authors have the option to publish the peer review history of their article (what does this mean?). If published, this will include your full peer review and any attached files.

Reviewer #2: No

---

## [Editor Report · Acceptance letter]

23 Feb 2022

PONE-D-21-31750R1 

Açai supplementation *(Euterpe oleracea Mart.)* attenuates cardiac remodeling after myocardial infarction in rats through different mechanistic pathways 

Dear Dr. Figueiredo:

I'm pleased to inform you that your manuscript has been deemed suitable for publication in PLOS ONE. Congratulations! Your manuscript is now with our production department. 

Kind regards, 

on behalf of

Dr. Luis Eduardo M Quintas 

Academic Editor

PLOS ONE